# (Pro)Renin Receptor Antagonism Attenuates High-Fat-Diet–Induced Hepatic Steatosis

**DOI:** 10.3390/biom13010142

**Published:** 2023-01-10

**Authors:** Ariana Julia B. Gayban, Lucas A. C. Souza, Silvana G. Cooper, Erick Regalado, Robert Kleemann, Yumei Feng Earley

**Affiliations:** 1Departments of Pharmacology and Physiology & Cell Biology, School of Medicine, University of Nevada, Reno, NV 89557, USA; 2Center for Molecular and Cellular Signaling in the Cardiovascular System, University of Nevada, Reno, NV 89557, USA; 3Department of Metabolic Health Research, The Netherlands Organization for Applied Scientific Research (TNO), 2333BE Leiden, The Netherlands

**Keywords:** (Pro)renin receptor, glycerol-3-phosphate acyltransferase 3, peroxisome proliferator activated receptor γ, NAFLD

## Abstract

Non-alcoholic fatty liver disease (NAFLD) comprises a spectrum of liver damage directly related to diabetes, obesity, and metabolic syndrome. The (pro)renin receptor (PRR) has recently been demonstrated to play a role in glucose and lipid metabolism. Here, we test the hypothesis that the PRR regulates the development of diet-induced hepatic steatosis and fibrosis. C57Bl/6J mice were fed a high-fat diet (HFD) or normal-fat diet (NFD) with matching calories for 6 weeks. An 8-week methionine choline-deficient (MCD) diet was used to induce fibrosis. Two weeks following diet treatment, mice were implanted with a subcutaneous osmotic pump delivering either the peptide PRR antagonist, PRO20, or scrambled peptide for 4 or 6 weeks. Mice fed a 6-week HFD exhibited increased liver lipid accumulation and liver triglyceride content compared with NFD-fed mice. Importantly, PRO20 treatment reduced hepatic lipid accumulation in HFD-fed mice without affecting body weight or blood glucose. Furthermore, PRR antagonism attenuated HFD-induced steatosis, particularly microvesicular steatosis. In the MCD diet model, the percentage of collagen area was reduced in PRO20-treated compared with control mice. PRO20 treatment also significantly decreased levels of liver alanine aminotransferase, an indicator of liver damage, in MCD-fed mice compared with controls. Mechanistically, we found that PRR antagonism prevented HFD-induced increases in PPARγ and glycerol-3-phosphate acyltransferase 3 expression in the liver. Taken together, our findings establish the involvement of the PRR in liver triglyceride synthesis and suggest the therapeutic potential of PRR antagonism for the treatment of liver steatosis and fibrosis in NAFLD.

## 1. Introduction

Non-alcoholic fatty liver disease (NAFLD), comprising a spectrum of pathological liver disease stages, including liver cell damage, is closely associated with diabetes, metabolic syndrome, and obesity. In light of the prevalence of obesity and the alarming growth in type 2 diabetes frequency, this condition presents a rising health threat in Western countries, where its incidence ranges from 20% to 30% and continues to grow, especially within the pediatric population [1]. Prominent hallmarks of this condition include an initial abnormal accumulation of triglycerides in hepatocytes resulting from, among other things, decreased β-oxidation, increased de novo lipogenesis, and increased free fatty acid flux from adipose tissue [2] in the absence of excessive alcohol intake. The pathogenesis of this disease is marked by a gradual loss in the ability of the liver to metabolize fatty acids and carbohydrates, leading to an abnormal accumulation of lipids in lipid droplets within hepatocytes (i.e., fatty liver). The recurrent instigation of hepatic steatosis is believed to increase the susceptibility of the liver to more severe forms of liver damage, typically in the form of non-alcoholic steatohepatitis (NASH), characterized by steatosis in conjunction with inflammation, which may present increasingly severe forms of hepatic fibrosis [3]. Progression of this disease is accompanied by irreversible liver damage and life-threatening complications, including hepatocellular carcinoma and, ultimately, liver failure. Current treatments for NAFLD are ineffective in preventing disease progression towards NASH and fibrosis, underscoring the need for the development of novel and effective treatment methods as well as the identification of valid drug targets.

One novel target that warrants investigating for the treatment of NAFLD is the (pro)renin receptor (PRR). The PRR, a single transmembrane receptor broadly expressed in the kidneys, heart, vascular smooth muscle, brain, adipose tissue, liver, and eye [4,5,6,7,8,9], is a key component of the renin-angiotensin system (RAS) that is involved in regulating blood pressure through angiotensin II-dependent and angiotensin II-independent pathways [10]. Reports have implicated prorenin-mediated PRR functions in various tissues in the development of metabolic syndrome features, including diabetes, obesity, and obesity-related hypertension [11,12,13]. In addition, several studies have identified the PRR as an emerging player in the regulation of lipid metabolism [8,14], suggesting that the PRR could be a potential novel target for the treatment of metabolic syndrome-related diseases, including non-alcoholic fatty liver.

In the current study, we used PRR antagonism, employing the 20-amino-acid peptide PRO20, to investigate the role of the PRR in modulating liver lipid metabolism in the setting of NAFLD. PRO20, which consists of the first 20 amino acids of the (pro)renin prosegment [15], displays high specificity for both the human and mouse PRR and outcompetes (pro)renin for PRR binding sites. The use of PRO20 as a tool for studying the action of PRR in the pathogenesis of disease has been validated previously in mouse models [15,16,17]. In the current study, we demonstrate the action of the PRR as a novel regulator of hepatic triglyceride metabolism and potential therapeutic tool against the development of fatty liver and NASH.

## 2. Materials and Methods

### 2.1. Animals

Male 16-week-old C57BL/6J mice were obtained from Jackson Laboratories (strain#: 000664). Mice were housed individually and fed either a high-fat diet (HFD; D12492, Research Diets Inc., New Brunswick, NJ, USA) or a normal-fat diet (NFD; D12450J, Research Diets Inc., New Brunswick, NJ, USA) containing 60% and 10% of kCal from fat, respectively, for 6 weeks. An 8-week methionine choline-deficient (MCD) diet (TD.90262, Envigo, Indianapolis, IN, USA) regimen was used to induce a NASH phenotype in 16-week-old male C57BL/6J mice under the same single-housed conditions. Body weight and food intake were measured weekly for all mice. Following the diet modification, mice were sacrificed in the fed state, and tissues were collected for molecular experiments. All animal procedures were approved by the Institutional Animal Care and Use Committee at the University of Nevada, Reno, and were performed in accordance with the National Institutes of Health Guidelines for the care and use of experimental animals.

### 2.2. Subcutaneous Administration of PRO20, Losartan, and Controls Using Osmotic Minipumps

After 2 weeks of diet treatment (NFD or HFD), mice received subcutaneous osmotic minipump implants (Alzet Micro-Osmotic Pump Model 1004, DURECT, Cupertino, CA, USA) administering the PRO20 peptide (LPTRTATFERIPLKKMPSVREI) or scrambled peptide control (LRTETPITMIPSAERVFRKKPL) at 700 μg/kg/d for the remaining 4 weeks of treatment. For the MCD diet, mice received a similar osmotic minipump implant (Alzet Micro-Osmotic Pump Model 2006, DURECT, Cupertino, CA, USA) after 2 weeks of MCD treatment that delivered the same dose of PRO20 or scrambled peptide over the remaining 6 weeks of diet treatment. Mice were anesthetized by isoflurane inhalation and then subcutaneously implanted with osmotic minipumps infusing PRO20 or scrambled (control) peptide. A separate study using losartan, an angiotensin II type 1a receptor (AT1aR) blocker, was performed using the aforementioned 6-week HFD and NFD diet treatment regimen. Specifically, after 2 weeks of diet treatment, mice received subcutaneous osmotic minipump implants that delivered either losartan (10 mg/kg/d) or 0.9% saline (control) for the subsequent 4 weeks. All animals were housed singly in standard forced-air shoebox cages and were maintained on their respective diets until the end of the treatment period.

### 2.3. Fasting Blood Glucose Measurements and Glucose Tolerance Tests

Fasting blood glucose (FBG) levels were measured in all mice at baseline and after 6 weeks of either a NFD or HFD regimen. Mice were fasted for 16 h (6 PM to 10 AM) in clean cages before glucose measurements. Blood was collected from each mouse by creating a 1-mm cut at the tip of the tail, and subsequent FBG values were measured using a Bayer 7393A Contour blood glucose meter. Glucose was measured in duplicate for each mouse, and the mean of the two values was taken as the final measurement.

Glucose tolerance tests (GTTs) were performed before the beginning of treatment and after 6 weeks of diet modification. Mice were fasted for 16 h (6 PM to 10 AM) in clean cages prior to beginning the GTT. After fasting, baseline blood glucose was measured and a solution of 10% glucose in 0.9% sterile saline (1 g/kg body weight) was injected intraperitoneally to elevate blood glucose levels. For GTTs, blood glucose was measured and recorded for each mouse 15, 30, 60, 90, and 120 min after injection of glucose using a Bayer blood glucose meter, as described above.

### 2.4. Oil Red O Staining

At the end of the 6-week study, all mice were sacrificed, and liver tissue was processed to obtain frozen liver cross-sections as well as paraffin-embedded cross-sections. The left lobe of the liver was kept in 4% paraformaldehyde for 24 h, then in 30% sucrose for the following 24 h, and subsequently frozen in tissue freezing medium (TFM-Y; General Data, Cincinnati, OH, USA). Serial frozen liver sections (10 µm thickness) were cut using a cryostat (Leica Biosystems, Deer Park, IL, USA), then attached to glass slides by air drying and fixed in formalin for 5 min. After a 1-min wash in tap water, slides were rinsed in 60% isopropanol and stained with Oil Red O solution (Abcam, ab150678) for 30 min. After staining, slides were rinsed in 60% isopropanol, counterstained with Mayer’s Hematoxylin (1 g/L; MHS32, Sigma Aldrich, St. Louis, MO, USA) for 10 s, and then mounted. A total of 297 images from an average of 5–10 images per mouse were captured using a light microscope (BZX-710; Keyence, Itasca, IL, USA), and the percent area of Oil Red O staining in each image was analyzed using ImageJ/FIJI. The average value from all images from each mouse was presented as an individual data point.

### 2.5. Hematoxylin & Eosin Staining

Hematoxylin & Eosin (H&E) staining was performed on paraffin-embedded median lobes of the liver. Paraffin-embedded liver tissues were cut at a thickness of 10 µm using a microtome (AccuCut Tissue-Tek; Sakura Finetek, Torrance, CA, USA). Slide-mounted tissues were rehydrated using a decreasing graded series of ethanol concentrations (100%, 95%, 80%, 70%), then stained with Harris Hematoxylin (6765003; ThermoFisher, Waltham, MA, USA) and Eosin Y (Sigma, H911032). Sections were then dehydrated with an increasing graded series of ethanol concentrations (50%, 70%, 80%, 95%, 100%) and mounted. A total of 170 images from an average of 5–10 images per mouse were acquired under a light microscope (BZX-710; Keyence, Itasca, IL, USA) and used for further analysis of liver grade and microvesicular and macrovesicular steatosis.

### 2.6. Assignment of Histological NAFLD Scores

A previously developed rodent NAFLD grading system [18] was used to score the extent of overall NAFLD development in each treatment group and the severity of hepatocellular vesicular steatosis (both microvesicular and macrovesicular steatosis). Microvesicular steatosis was defined as the presence of hepatocellular lipid vacuoles that did not displace the nucleus to the side, whereas macrovesicular steatosis featured the presence of large lipid vacuoles that displaced the nucleus. The severity of both types of vesicular steatosis was based on the total area of the slide affected, determined from an analysis of images taken at 20× magnification. Scores were assigned from least to most severe based on the percentage of area affected, as follows: 0, <5%; 1, 5–33%; 2, 34–66%; and 3, >66%. The sum of the scores derived from both microvesicular and macrovesicular area percentages was considered the total steatosis grade.

### 2.7. Picrosirius Red Staining

Paraffin-embedded sections of the median lobe of the liver were cut at a thickness of 10 µm and then stained with Picrosirius Red, which supplements Masson’s Trichrome staining in assessing hepatic fibrosis. Slide-mounted sections were incubated in xylene and then rehydrated by soaking in a descending graded series of ethanol concentrations (100%, 95%, 80%, 70%). Samples were then washed in phosphate-buffered saline (PBS) for 3 min, followed by an 8-min incubation in Weigert’s hematoxylin and 10-min wash in tap water. The tissue was then stained in Sirius Red solution for 1 h, after which slides were immersed in two changes of acidified water for 2 min each. Samples were then dehydrated in 3 changes of 100% ethanol for 2 min each, cleared in xylene for 5 min, and then mounted. A total of 182 images from an average of 5–10 images per mouse were captured using a light microscope (BZX-710; Keyence, Itasca, IL, USA), and the area of red collagen staining, expressed as a percentage, was measured using ImageJ/FIJI. The average value from all images from each mouse was presented as an individual data point.

### 2.8. Liver Alanine and Aspartate Aminotransferase Activity Assay

Liver injury in MCD-treated mice was assessed biochemically by quantifying plasma alanine aminotransferase (AST) and aspartate aminotransferase (ALT) activity. Colorimetric assays for AST (MAK055; Sigma-Aldrich, St. Louis, MO, USA) and ALT (MAK052; Sigma-Aldrich, St. Louis, MO, USA) activity levels were performed according to the manufacturer’s protocols. Briefly, plasma samples were added to microplate wells containing prepared AST or ALT reaction mixes. The optical density (OD) of samples was measured at a wavelength of 450 nm for AST and 570 nm for ALT using a microplate reader (FlexStation 3; Molecular Devices, San Jose, CA, USA). Plates were repeatedly incubated at 37 °C for 5 min, and OD measurements were taken following each incubation until the value of the most active sample eclipsed the value of the highest standard. AST and ALT activity was calculated in milliunits/mL using a standard curve generated for each assay, assay reaction time, and the difference between the final and initial OD measurement.

### 2.9. Quantitative Reverse Transcription-Polymerase Chain Reaction (RT-qPCR)

Total mRNA was extracted from frozen liver samples using the Trizol solubilization and extraction method according to the manufacturer’s protocol (15596018; ThermoFisher, Waltham, MA, USA). The isolated RNA was resuspended in 50 µL of ultra-pure water, and the quality and yield of total RNA were determined using a Nanodrop spectrophotometer.

RNA contaminants were eliminated using a DNAse I treatment kit (BP81071; ThermoFisher, Waltham, MA, USA), and cDNA was produced from total RNA by reverse transcription using a High-Capacity cDNA reverse transcription kit (4368814; ThermoFisher, Waltham, MA, USA). Real-time qPCR was performed on a Quantstudio 3 System in 20-µL reactions using SYBR Green PCR Master Mix (ThermoFisher, Waltham, MA, USA). The mRNA levels of the following targets were measured and reported as fold-change in mRNA expression, determined using the ΔΔCT method: peroxisome proliferator activated receptor (PPAR)-α, -β, and -γ; β-actin; carbohydrate-responsive element-binding protein (CHREBP); sterol regulatory element-binding protein 1c (SREBP1c); acetyl-CoA carboxylase (ACC); fatty acid synthase (FAS); ATP-citrate lyase (ATPCL); and glycerol-3-phosphate acyltransferase 3 (GPAT3). The primer sequences used for each gene are listed in Table 1.

### 2.10. Western Blot Analysis

Protein abundance of β-actin and GPAT3 was investigated by Western blot analysis. Liver total protein was extracted using RIPA lysis buffer (89901; ThermoFisher, Waltham, MA, USA), and protein concentrations were determined using BCA assays (23225; ThermoFisher, Waltham, MA, USA). Protein-containing samples were prepared for sodium dodecyl sulfate-polyacrylamide gel electrophoresis (SDS-PAGE) by denaturation in a solution containing 2% SDS and 100 mM DTT at 95 °C for 5 min, followed by centrifugation at 13,000× *g* for 10 min. Equal amounts of protein (10–30 µg) were resolved on 4–12% Tris-glycine gels ( NW04125BOX; Invitrogen, Waltham, MA, USA). After gel electrophoresis, proteins were transferred onto a nitrocellulose membrane for 70 min at 4 °C using a traditional wet transfer method. Blots were then blocked by incubating in Tris-buffered saline containing 2.5% BSA for 1 h at room temperature. After blocking, blots were incubated with rabbit anti-mouse β-actin (1:1000 dilution; 8457S Cell Signaling Technology, Danvers, MA, USA) or rabbit anti-mouse GPAT3 (1:500 dilution; 20603-1-AP; ThermoFisher, Waltham, MA, USA) primary antibody. Following incubation with primary antibody, blots were transferred to a solution containing horseradish peroxidase (HRP)-linked anti-rabbit secondary IgG antibody (1:1000 dilution; 7074S; Cell Signaling Technology, Danvers, MA, USA) and incubated for 1 h at room temperature. Immunoreactive proteins were detected using SuperSignal West Dura Extended Duration Substrate (34076; ThermoFisher, Waltham, MA, USA) in conjunction with the BioRad ChemiDoc MP Imaging System. Band intensity was quantified using ImageJ/FIJI software.

### 2.11. Statistial Analysis

Normality was assessed using a Shapiro-Wilk test. Statistical analyses were performed using Graphpad Prism 9.0 software without removal of outliers. Data are expressed as means ± SEM, and were analyzed by Student’s t-test, one-way analysis of variance (ANOVA), or two-way ANOVA with Fisher’s LSD test or Bonferroni’s post hoc tests to correct for multiple comparisons, as appropriate. Statistical comparisons were performed using GraphPad Prism 9 software (GraphPad Software, La Jola, CA, USA). Differences with *p*-values < 0.05 were considered statistically significant.

## 3. Results

### 3.1. PRR Antagonism Attenuates HFD-Induced Lipid Accumulation in the Liver

Previous studies have shown that HFD induces hepatic steatosis in as few as 2 weeks [19,20,21]. To determine whether the PRR regulates hepatic lipid metabolism and plays a role in reversing or attenuating the development of hepatic steatosis, we used the PRR antagonist, PRO20, to block PRR activation in an HFD-induced mouse model of NAFLD after 2 weeks of HFD. PRR antagonism reduced fat accumulation in the liver, as detected by Oil Red O staining (Figure 1A). Quantitative analyses showed that, after a period of 6 weeks, HFD induced a significant accumulation of fat (16.6% ± 1.0%) compared with NFD controls (3.3% ± 0.4%, *p* < 0.00001). This effect of HFD was attenuated by administration of PRO20, which significantly decreased hepatic fat accumulation (7.7% ± 1.4%, *p* < 0.00001) compared with HFD controls receiving scrambled peptide (Sc Peptide) (Figure 1B), albeit without completely normalizing fat content compared with mice that received NFD and scrambled peptide treatment. PRO20 had no effect on lipid accumulation in NFD-fed mice.

To further confirm the lipid-accumulation status in the liver, we measured total hepatic triglyceride levels, normalized to total protein present in each sample. As shown in Figure 1C, 6 weeks of HFD feeding induced a significant increase in total hepatic triglycerides (72.9 ± 14.2 mg/g protein) in mice that received the scrambled peptide. Notably, this effect was blunted by administration of PRO20, which reduced the concentration of hepatic triglycerides by nearly 72% (to 20.5 ± 6.0 mg/g protein) compared with HFD controls. These data confirm Oil Red O results showing that HFD treatment induces lipid accumulation and that PRR antagonism alleviates this accumulation. The increase in hepatic triglycerides in HFD-fed mice was accompanied by a significant increase in plasma triglyceride levels (Figure 1D). Specifically, circulating triglycerides increased from 134.9 ± 13.3 mg/dL in NFD-fed mice to 213.1 ± 19.8 mg/dL in mice fed a HFD for 6 weeks (*p* = 0.0137). Although plasma triglyceride levels trended lower in HFD-fed mice treated with PRO20 (165.6 ± 13.94 mg/dL), this decrease did not achieve statistical significance (*p* = 0.1123).

### 3.2. Subcutaneous Infusion of Losartan Does Not Reduce Hepatic Lipid Accumulation in HFD-Fed Mice

Activation of the PRR mediates formation of angiotensin (Ang) peptides as well as stimulation of Ang II-independent signaling pathways [10,22]. To investigate the extent to which Ang II/AT_1_R signaling activation impacts the development of non-alcoholic fatty liver, we administered the AT_1_R antagonist, losartan, using the same protocol as above for PRO20. Representative images of Oil Red O-stained tissues from NFD- or HFD-fed mice treated with losartan or 0.9% saline are shown in Figure 1E. As expected, 6 weeks of HFD induced an increase in liver lipid accumulation (20.0% ± 1.6%) relative to NFD-fed mice infused with saline (4.3% ± 0.6%, *p* < 0.0001). Hepatic lipid accumulation tended to be lower with subcutaneous losartan infusion for 4 weeks (16.5% ± 0.5%), but the Oil Red O-positive area was not significantly reduced compared with controls (20.0% ± 1.6%, *p* = 0.09). Thus, at the end of the 6-week HFD regimen (Figure 1F), HFD-fed mice that received losartan treatment still displayed an increase in hepatic lipid content compared with NFD-fed control mice (4.3% ± 0.6%, *p* < 0.0001). These data indicate that losartan does not attenuate HFD-induced lipid accumulation in the liver in our experimental time frame.

### 3.3. PRO20 Attenuates HFD-Induced Hepatic Steatosis

To examine structural changes in the liver following HFD and compare the extent of histological changes among groups, we performed H&E staining (Figure 2A), applying a histological scoring system commonly used to gauge the severity of NAFLD. Using a 0–3 scale developed for liver grading in rodents described by Liang et al. [18], we found that HFD-fed mice scored consistently higher than their NFD-fed counterparts in overall NAFLD severity (Figure 2B). In contrast, the liver grading scores of HFD-fed mice administered PRO20 were significantly lower than those of HFD controls and on par with those of NFD-fed mice.

Next, we evaluated characteristic histological features of fatty liver disease, focusing on the development of microvesicular and macrovesicular steatosis [18]. As shown in Figure 2C, HFD induced an increase in microvesicular (red circles) and macrovesicular (black circles) steatosis as a result of increased hepatic lipid, with the area occupied by microvesicular steatosis reaching 46.5% ± 8.0% in HFD-fed mice compared with 11.4% ± 4.6% in NFD controls (*p* = 0.0013). Notably, we found a significant reduction in the severity of microvesicular steatosis in HFD-fed mice following treatment with the PRR antagonist PRO20, which normalized the total steatosis area, reducing it to 10.6% ± 4.7% (*p* = 0.0007) (Figure 2D). Six weeks of HFD did not significantly increase macrovesicular steatosis compared with that observed in NFD-fed mice, and there were no differences in macrovesicular steatosis among treatment groups (Figure 2E).

### 3.4. PRO20 Decreases Hepatic Fibrosis Development and Liver Injury in MCD Diet-Fed Mice

To investigate the role of the PRR under more severe hepatic fibrosis conditions, we employed an MCD diet model and administered PRO20 and scrambled peptides in tandem, as depicted in the protocol shown in Figure 3A. Body weight and food intake parameters were monitored weekly in all mice following the start of the MCD diet. No significant differences in body weight or food intake were observed between PRO20 and scrambled peptide treatment groups fed an MCD diet (Figure 3B). However, mice in both groups experienced a large drop in body weight, which is characteristic of this diet model [23]. Following an 8-week MCD diet and infusion of PRO20 or scrambled peptide for 6 weeks, livers were assessed for collagen deposition indicative of fibrosis development using Picrosirius Red staining and a biochemical collagen assay. Representative images (Figure 3C) and quantification of Picrosirius Red-stained area (Figure 3D) showed that PRO20 significantly reduced collagen deposition following MCD treatment, reducing the total area to 6.0% ± 0.6% compared with 8.9% ± 0.2% for scrambled peptide (*p* < 0.0001), indicating that the PRR may regulate the onset of liver fibrosis in MCD diet-induced NASH. Plasma ALT and AST levels, markers of liver injury, were also assessed in both PRO20- and scrambled peptide-treated mice (Figure 3E,F). PRO20 treatment significantly decreased circulating ALT activity (422.3 ± 65.2 U/L) compared with scrambled peptide controls (683.1 ± 83.41 U/L, *p* = 0.0249), but had no effect on AST activity.

### 3.5. PRR Antagonism Attenuates HFD-Induced Elevation of PPARγ and GPAT3 in the Liver

Because HFD-fed, PRO20-treated mice displayed significantly reduced hepatic triglyceride levels, we investigated whether genes involved in de novo lipogenesis in the liver were affected by the PRR antagonist PRO20. A 6-week HFD regimen, with or without PRO20, had no effect on mRNA expression levels of transcription factors involved in lipogenesis, including liver X receptor alpha (LXRα), carbohydrate response element binding protein (ChREBP), and sterol regulatory element-binding protein 1 (SREBP1c) (Figure 4A–C). As part of our investigation of changes in lipid homeostasis, we also assessed hepatic mRNA expression of members of the PPAR family of transcriptional regulators. The three known PPAR isoforms—PPARα, PPARδ, and PPARγ—have been shown to play significant roles in NAFLD pathology and liver physiology. In particular, activation of PPARα has been shown to promote fatty acid oxidation and transport, whereas the action of PPARγ leads to increased hepatic lipogenesis, triglyceride storage, and adipogenesis [24,25]. PPARδ, though most abundant in muscle tissue, functions through interactions with the other PPAR isoforms to inhibit hepatic lipogenesis and insulin resistance [26,27]. After a 6-week HFD, hepatic PPARγ was significantly upregulated (2.6 ± 0.6) compared with that observed in NFD controls (1.13 ± 0.19, *p* = 0.0037) (Figure 4D), an effect that was attenuated by administration of PRO20. Hepatic PPARα mRNA expression also trended higher in HFD-fed mice; although this increase did not reach statistical significance, it was significantly attenuated by PRO20 treatment (1.4 ± 0.1) compared to treatment with scrambled peptide (0.8 ± 0.1, *p* = 0.0040) (Figure 4E). PRO20 treatment also showed a tendency to reduce PPARδ mRNA levels in HFD-fed mice (1.0 ± 0.1) compared to HFD-fed mice treated with scrambled peptide (0.74 ± 0.07, *p* = 0.0319), although no change in PPARδ expression was found between HFD- and NFD-fed scramble peptide mice (Figure 4F).

Changes in the expression of mRNAs encoding the corresponding downstream lipogenic enzymes, regulated by the above-described transcription factors, were investigated following HFD and NFD treatments. No changes in hepatic expression of transcripts for fatty acid synthase (*Fas*), ATP citrate lyase (*Atpcl*), or acetyl-CoA carboxylase (*Acc*) enzymes were detected (Figure 5A–C). However, hepatic *Gpat3* (2.9 ± 0.5) was increased after a 6-week HFD compared with NFD feeding (0.9 ± 0.1, *p* = 0.0001) (Figure 5D). Notably, this increase in hepatic *Gpat3* mRNA expression was significantly attenuated in HFD-fed mice administered PRO20 (1.84 ± 0.18, *p* = 0.0183) compared with their control HFD counterparts. Western blotting assays confirmed modulation of GPAT3 protein by PRR antagonism, revealing that hepatic GPAT3 protein increased in HFD-fed mice (1.53 ± 0.03) compared with NFD controls (1.00 ± 0.18, *p* = 0.0242) (Figure 5E) and that this increase was absent in HFD-fed mice treated with PRO20 (0.93 ± 0.19, *p* = 0.0127) (Figure 5F). These data suggest that the PRR may be involved in lipogenesis pathways mediated by PPARγ and GPAT3.

### 3.6. PRR Antagonism Has No Effect on Body Weight or Glucose Homeostasis

To investigate whether PRR antagonism impacts obesity or glucose homeostasis following HFD feeding, we monitored body weights of all mice on a weekly basis (Figure 6A). PRO20 treatment did not affect body weight of NFD-fed mice compared with that of scrambled peptide-treated, NFD-fed mice. After a 6-week HFD, both scrambled peptide- and PRO20-treated mice exhibited a significant increase in body weight compared with their NFD counterparts. However, no difference in body weight was observed between PRO20- and scrambled peptide-treated mice fed a 6-week HFD (Figure 6A). Similarly, there was no difference in food or caloric intake between PRO20- and scrambled peptide-treated mice fed a 6-week HFD (Figure 6B,C). These data indicate that PRR antagonism has no effect on body weight, food intake, or caloric consumption.

At baseline, there was no difference in fasting blood glucose (FBG) or glucose handling among groups, as demonstrated by fasting blood glucose and glucose tolerance tests (GTTs) before treatment (Figure 6D–F). In contrast, at the end of the diet regimen, FBG was significantly elevated in both scrambled peptide-treated (131.4 ± 8.8 mg/dL, *p* = 0.0021) and PRO20-treated (134.9 ± 6.9 mg/dL, *p* = 0.0007), HFD-fed mice compared with their corresponding NFD-fed controls (Figure 6G). NFD-fed mice displayed no change in FBG over the duration of treatment. A GTT performed at the end of the 6-week dietary regimen (NFD or HFD) to assess the state of glucose handling (Figure 6H,I) showed that the HFD significantly impaired glucose tolerance (15,169 ± 786 RU, *p* = 0.0355), as reflected in an AUC analysis of blood glucose values (Figure 6I). We found no difference in glucose tolerance between HFD-fed mice treated with a scrambled peptide and those treated with PRO20 after 6 weeks. These data suggest that 6 weeks of HFD impairs glucose metabolism and promotes body weight gain, and that PRR antagonism has a minimal impact on these parameters.

## 4. Discussion

The RAS has been implicated in the pathogenesis of NAFLD, and research has demonstrated a role for RAS intermediates in hepatic glucose metabolism, lipid processing, and insulin sensitivity [28,29,30]. The PRR in particular has been shown to be involved in hepatic cholesterol clearance; however, the mechanisms through which the PRR regulates other aspects of lipid metabolism in the context of fatty liver and NASH development have remained elusive. In this study, we demonstrate that PRR antagonism in vivo ameliorates the development of HFD-induced fatty liver disease in mice. Among our key findings was the demonstration that PRO20 reduces HFD-induced lipid deposition in the liver as well as liver triglyceride content without significantly impacting body weight or glucose metabolism. We also show that losartan, an AT_1_R blocker, does not attenuate development of hepatic steatosis in HFD-fed mice, indicating that the effect of PRR antagonism in hepatic steatosis is probably RAS-independent. A further mechanistic investigation into whether modulation of the de novo lipogenesis pathway underlies the protective effect of PRR antagonism against lipid accumulation revealed that PRO20 treatment reduced mRNA expression of the transcription factor PPARγ and its downstream target, *Gpat3*, encoding the lipogenesis pathway enzyme GPAT3, in HFD-fed mice. Lastly, a role for the PRR in the development of fibrosis was demonstrated in MCD diet-fed mice administered PRO20, which caused a reduction in hepatic collagen deposition and serum ALT levels.

One surprise finding in this study was that losartan did not significantly reduce hepatic lipid accumulation following HFD. Previous research has shown that the PRR contributes to the regulation of cholesterol metabolism, and that silencing the hepatic PRR decreases low-density lipoprotein receptor (LDLR) and SORT1 protein levels in a RAS-independent manner, thereby reducing LDL clearance in the liver [8,31]. It has also been established that hepatic PRR inhibition reduces triglyceride content, a finding also demonstrated in the current study. To narrow the mechanistic possibilities for the action of PRR inhibition on steatosis development, we performed the losartan study using a diet treatment identical to that employed for the PRO20 treatment protocol. Losartan is an AT_1_R antagonist that competitively blocks the action of Ang II at AT_1_Rs at the dose employed in this study, leading to a decrease in blood pressure [32,33,34]. Administration of losartan has been shown to attenuate hepatic steatosis in some animal models; however, in other cases, no effect of this intervention on steatosis development, weight gain, or glucose handling was reported [35,36]. Nevertheless, in the current study, losartan treatment did not reduce fat accumulation in the liver following a 6-week HFD. Accordingly, modulation of RAS-independent signaling pathways in hepatic lipid metabolism likely accounts for the effects of the PRR antagonist, PRO20, in ameliorating the development of fatty liver.

The lack of difference in body weight gain and glucose handling between HFD-fed mice treated with scrambled peptide versus PRO20 suggests that the PRR may target pathways separate from carbohydrate metabolism. Glucose metabolism and insulin sensitivity are key factors in the pathogenesis of NAFLD, and insulin signaling is involved in regulating fatty acid metabolism through upregulation of genes involved in de novo lipogenesis and downregulation of lipid-degradation pathway genes [37]. Notably, obesity coupled with insulin resistance and type 2 diabetes are common comorbidities that exacerbate the progression of NAFLD in a clinical setting. In the insulin-resistant state, hepatic glucose production and fatty acid uptake are elevated, leading to increased substrates for triglyceride synthesis [38]. The fact that PRR antagonism was unable to alter body weight, fasting blood glucose, or glucose handling in HFD-fed mice may suggest that our intervention only targets enzymatic pathways directly involved in lipid synthesis while leaving other clinical aspects of NAFLD pathology unchanged.

In HFD-induced non-alcoholic fatty liver development, PRR antagonism was associated with changes in the triglyceride synthesis pathway via PPARγ and its downstream target, GPAT3. PPARγ is a transcription factor found primarily in adipose and liver tissue that modulates a multitude of target genes involved in processes such as fat storage and import, insulin sensitivity, and inflammatory response [39]. In hepatocytes, PPARγ plays a critical role in lipid homeostasis by upregulating genes involved in de novo lipogenesis, fat uptake, and formation of lipid droplets [39,40,41]. A significant increase in hepatic PPARγ expression is also a common phenotype in NAFLD models and is associated with increased steatosis [42]. Interestingly, a recent study reported that the PRR gene is a target of PPARγ in vitro, providing a basis for functional interactions between the PRR and PPARγ [8]. Another confirmed target of activated PPARγ is the gene encoding GPAT3, one of four GPAT isoforms that catalyze the conversion of glycerol-3-phosphate and long-chain acyl-CoA to lysophosphatidic acid–the pivotal rate-limiting step in the de novo synthesis of triglycerides in mammals [43]. GPAT3 activation in response to PPARγ has been demonstrated extensively in white adipose tissue using PPARγ agonists [44]. Previous studies have shown that downregulation of GPAT3 and other triglyceride synthetic enzymes has a beneficial effect on HFD-induced hepatic lipid accumulation [45], underscoring the potential contribution of hepatic GPAT3 function to NAFLD. While the origin of excessive fat accumulation in the form of triglycerides in the liver varies, it has been estimated that nearly 30% of the hepatic triglycerides that accumulate during NAFLD development originate from de novo lipogenesis in the liver [2], suggesting that decreasing the extent of de novo lipogenesis in the liver is a viable therapeutic options for the treatment of NAFLD.

Finally, we investigated a potential role of the PRR in the development of NASH, a more severe stage of NAFLD, finding that PRR antagonism reduced fibrosis development in the MCD diet-induced NASH mouse model. Serum ALT levels were also reduced in MCD-fed mice treated with PRO20, indicating that PRR antagonism can decrease the severity of NASH-associated liver damage to a certain extent. ALT is commonly released from hepatocytes in response to lipid infiltrates, and subsequent cellular dysfunction and circulating ALT levels are considered indicative of liver damage [46]. However, the exact mechanisms through which the PRR acts on fibrogenic pathways remain unclear.

We acknowledge that the approach utilized in this current study has its limitations. All experiments were performed using only adult male mice; thus, our findings are most relevant to the pathology of fatty liver in males. A 6-week HFD regimen was found to be sufficient to produce the fatty liver phenotype in male C57BL/6J mice, with 4 weeks of PRO20 intervention being able to rescue this phenotype. However, whether PRO20 would produce similar effects in other steatosis models, including high-fructose–induced steatosis, remains to be examined in future studies. In addition, the mechanism by which PRR antagonism affects liver lipid regulation was investigated only in the context of changes in the hepatic lipogenesis pathway. Other pathways, including fatty acid β-oxidation and lipid transport (through VLDL secretion) [47] could also contribute to PRR antagonism-dependent alterations in the lipid metabolism profile of the liver. Moreover, because the PRR antagonist PRO20 was administered globally via an osmotic minipump over a set period of time, its effect on hepatic lipid metabolism could also be linked to crosstalk among other organs, particularly white adipose tissue. Additional experiments to validate the action of the PRR in the liver and confirm local tissue-specific effects may require the use of liver-specific targeting.

## 5. Conclusions

In conclusion, the PRR antagonist, PRO20, ameliorates HFD-induced fatty liver development in mice, whereas losartan, an AT_1_R blocker, does not attenuate the development of hepatic steatosis in HFD-fed mice, suggesting that the effect of PRR antagonism in hepatic steatosis is RAS-independent. Mechanistically, PRO20 reduced PPARγ and expression of its downstream target *Gpat3*, encoding the de novo lipogenesis pathway enzyme, GPAT3, in HFD-fed mice. Lastly, PRR antagonism reduced hepatic collagen deposition and improved liver function in a mouse model of MCD diet-induced liver fibrosis. Our results support the hypothesis that the PRR is involved in de novo triglyceride synthesis pathways in the liver and the progression of hepatic fibrogenesis in NASH, strengthening the relevance of the PRR as a potential therapeutic target for treatment of the NAFLD spectrum. Future studies will be aimed at elucidating the actions of the PRR in hepatic lipid degradation and transport, as well as in regulating NASH pathology and associated fibrogenesis.

## Figures and Tables

**Figure 1 biomolecules-13-00142-f001:**
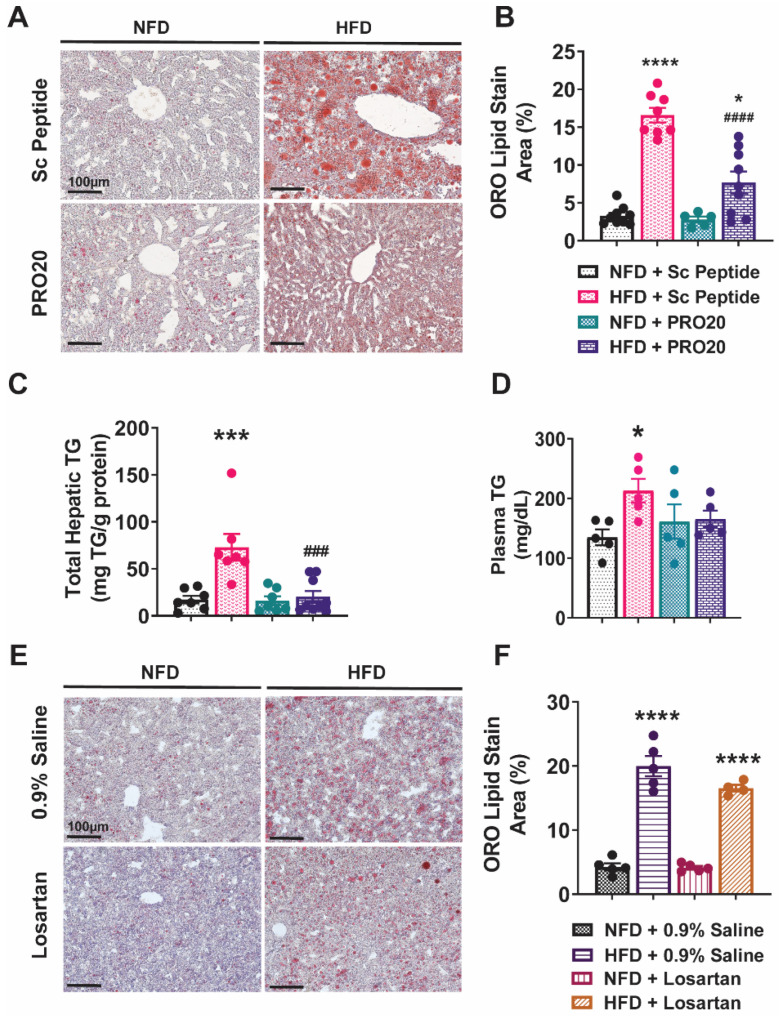
The PRR antagonist, PRO20, but not losartan, reduces hepatic steatosis under HFD conditions. (**A**) Representative images of hepatic lipid accumulation, visualized by Oil Red O (ORO) staining, in mice from PRO20 and scrambled peptide control groups fed either a HFD or NFD. (**B**) Quantification of ORO staining, presented as the percentage area of red staining relative to total image area. (**C**) Biochemical assay of total hepatic triglycerides (TG) normalized to total liver protein levels. (**D**) Biochemical assessment of plasma TG concentrations in mice. (**E**) Representative images of hepatic lipid accumulation, determined by ORO staining, in mice from losartan and control groups under HFD or NFD conditions. (**F**) Quantification of ORO staining from the losartan study, presented as the area of red staining relative to the total image area, expressed as a percentage. Data are presented as means ± SEM (* *p* < 0.05, *** *p* < 0.001, **** *p* < 0.0001 vs. NFD + Sc Peptide or NFD + 0.9% Saline; ^###^
*p* < 0.001, ^####^
*p* < 0.0001 vs. HFD + Sc Peptide; one-way ANOVA); *n* = 5–10 mice/group.

**Figure 2 biomolecules-13-00142-f002:**
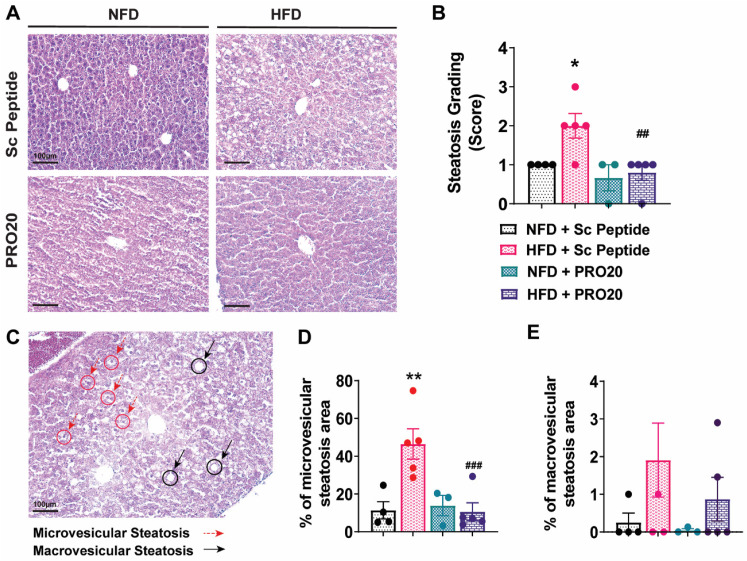
The PRR antagonist, PRO20, decreases hepatic steatosis severity and microvesicular steatosis abundance. (**A**) Representative images of H&E-stained hepatic tissue, showing differences in tissue structure among treatment groups. (**B**) Distribution of liver steatosis grades in mice. Scores (0–3) were assigned based on the presence of microvesicular and macrovesicular steatosis, with 0 being the least severe and 3 corresponding to maximum steatosis severity. (**C**) Examples of microvesicular steatosis (dashed red arrows) and macrovesicular steatosis (solid black arrows) in mouse liver tissue. (**D**,**E**) Quantification of microvesicular (**D**) and macrovesicular (**E**) steatosis area, reported as microvesicular/macrovesicular lipid area relative to total image area, expressed as a percentage (*n* = 3–5 mice/group). Data are presented as means ± SEM (* *p* < 0.05, ** *p* < 0.01 vs. NFD + Sc Peptide; ^##^
*p* < 0.01, ^###^
*p* < 0.001 vs. HFD + Sc Peptide; one-way ANOVA); *n* = 3–5 mice/group.

**Figure 3 biomolecules-13-00142-f003:**
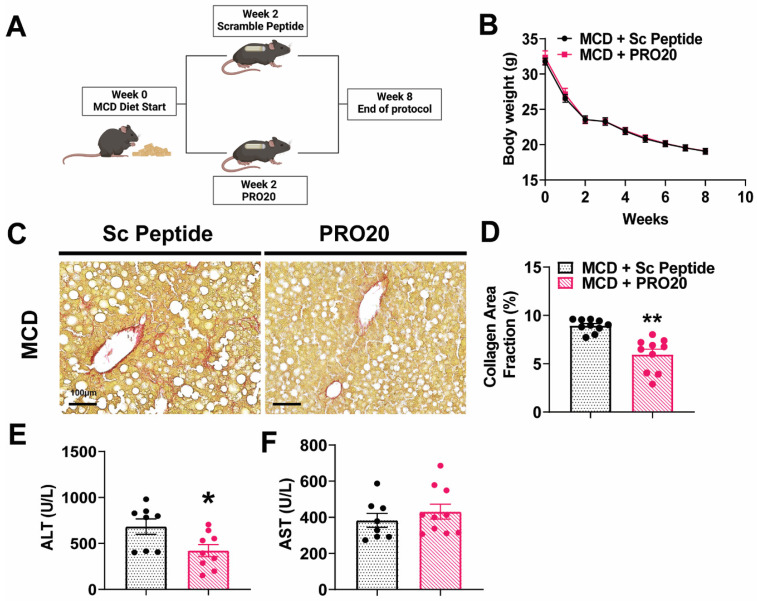
PRO20 treatment reduces hepatic fibrosis development and liver damage following 8 weeks of MCD Diet. (**A**) Schematic of MCD diet study protocol. (**B**) Summary data showing weekly body weight measurements in mice under MCD diet conditions. (**C**) Representative images of hepatic collagen deposition, visualized by Picrosirius Red staining. (**D**) Quantification of Picrosirius Red collagen stain. Data are presented as red collagen staining relative to total image area, expressed as a percentage. (**E**,**F**) Biochemical assessment of plasm ALT (**E**) and AST (**F**) levels. Data are presented as means ± SEM (* *p* < 0.05, ** *p* < 0.01 vs. MCD + Sc Peptide; Unpaired *t*-test); *n* = 8–10 mice/group.

**Figure 4 biomolecules-13-00142-f004:**
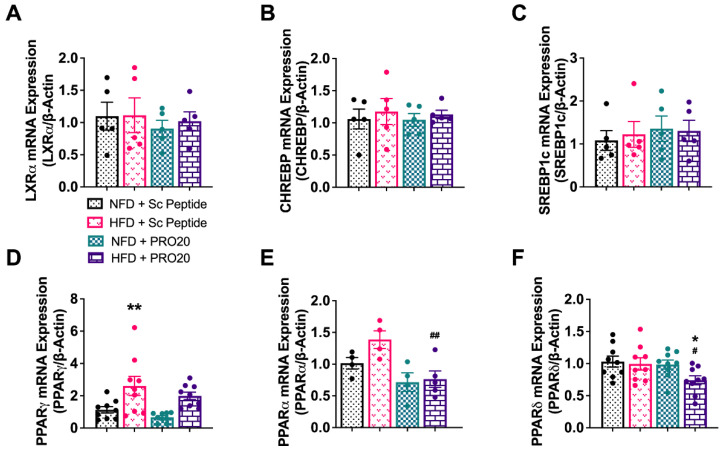
Six weeks of HFD increases hepatic mRNA expression of lipogenic PPARγ. Hepatic mRNA levels of the transcription factors LXRα (**A**), CHREBP (**B**), SREBP1c (**C**), PPARγ (**D**), PPARα (**E**), and PPARδ (**F**) following 6 weeks of HFD or NFD and either PRO20 or Sc Peptide treatment. The mRNA expression was assessed by RT-qPCR and was normalized to β-actin. Data are presented as means ± SEM (* *p* < 0.05, ** *p* < 0.01 vs. NFD + Sc Peptide; ^#^
*p* < 0.05, ^##^
*p* < 0.01 vs. HFD + Sc Peptide; one-way ANOVA); *n* = 4–10 mice/group.

**Figure 5 biomolecules-13-00142-f005:**
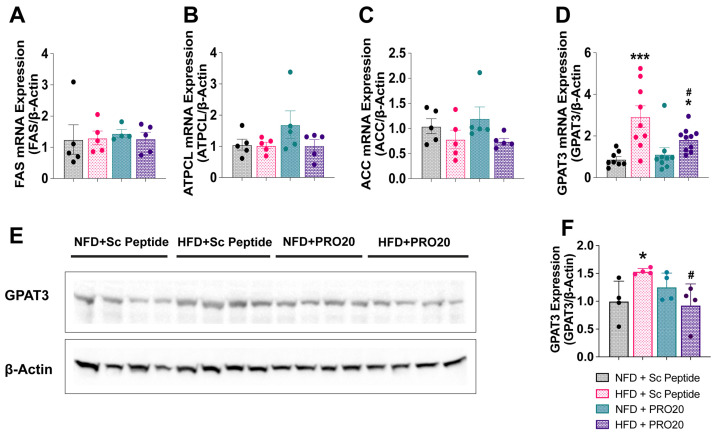
PRR antagonism prevents the increase in hepatic GPAT3 induced by 6 weeks of HFD. Hepatic expression of mRNAs encoding the enzymatic intermediates involved in triglyceride synthesis, FAS (**A**), ATPCL (**B**), ACC (**C**), and GPAT3 (**D**). (**E**) Western blot assessment of hepatic GPAT3 protein abundance. (**F**) Quantification of hepatic GPAT3 protein levels following 6 weeks of HFD or NFD and administration of either PRO20 or Sc Peptide. Data are presented as means ± SEM (* *p* < 0.05, *** *p* < 0.001 vs. NFD + Sc Peptide; ^#^
*p* < 0.05 vs. HFD + Sc Peptide; one-way ANOVA); *n* = 4–10 mice/group.

**Figure 6 biomolecules-13-00142-f006:**
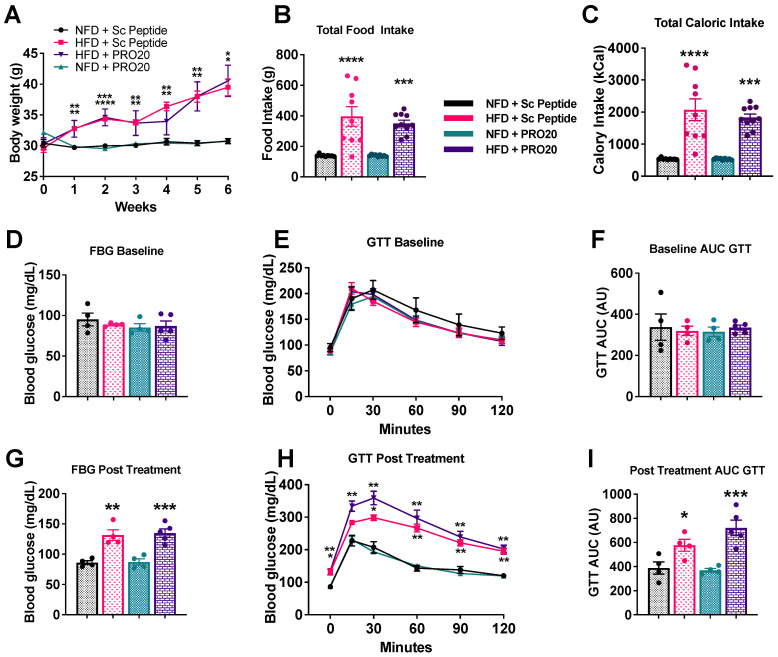
PRO20 treatment has no effect on body weight or glucose handling after 6 weeks of HFD. (**A**) Weekly body weight measurements in mice over the course of the study. (**B**) Total food intake over the 6-week protocol. (**C**) Total caloric intake over 6 weeks. (**D**–**F**): Baseline fasting blood glucose (FBG) levels (**D**), glucose tolerance test (GTT) and AUC analysis (**E**,**F**) before diet and PRR antagonism treatment. (**G**–**I**): FBG levels (**G**), GTT (**H**) and AUC analysis of 6 weeks post-treatment (**I**). Data are presented as means ± SEM (* *p* < 0.05, ** *p* < 0.01, *** *p* < 0.001, **** *p* < 0.0001 vs. NFD + Sc two-way ANOVA); *n* = 4–5 mice/group.

**Table 1 biomolecules-13-00142-t001:** Sequences of primers used in the qPCR analysis of PPARs and triglyceride synthesis components.

Gene	Primer Sequence (5’-3’)
β-actin	FWD: CCAGCCTTCCTTCTTGGGTA REV: AGAGGTCTTTACGGATGTCAACG
Peroxisome proliferator-activated receptor alpha (PPARα)	FWD: GTTCACGCATGTGAAGGCTG REV: GCGAATTGCATTGTGTGACATC
Peroxisome proliferator-activated receptor delta (PPARδ)	FWD: GCTCGAGTATGAGAAGTGCGA REV: CGGATAGCGTTGTGCGACAT
Peroxisome proliferator-activated receptor gamma (PPARγ)	FWD: GCTTGTGAAGGATGCAAGGGTTT REV: ATCCGCCCAAACCTGATGG
Fatty acid synthase (FAS)	FWD: CTGACTCGGCTACTGACACG REV: AATGGGGTGCACAAGGAACA
Acetyl-CoA carboxylase alpha (ACC)	FWD: GCCTTTCACATGAGATCCAGC REV: CTGCAATACCATTGTTGGCGA
ATP citrate lyase (ATPCL)	FWD: CCCAAGATTCAGTCCCAAGTC REV: TTGTGATCCCCAGTGAAAGG
Liver X receptor alpha (LXRa)	FWD: CTCTGCAATCGAGGTCATGCT REV: CAGCTCATTCATGGCTCTGGA
Sterol regulatory element-binding protein 2 (SREBP-2)	FWD: CTTCGAAGGCTGGCCCATA REV: AGGTGTCTACCTCTCCATGCTT
Sterol regulatory element-binding protein 1c (SREBP-1c)	FWD: GGAGCCATGGATTGCACATT REV: GGCCCGGGAAGTCACTGT
Carbohydrate response element binding protein (CHREBP)	FWD: AGTGCTTGAGCCTGGCCTAC REV: TTGTTCAGGCGGATCTTGTC
Glycerol-3-phosphate acyltransferase 3 (GPAT3)	FWD: AGCTTTGAAATCGGAGGAACC REV: AACTGCGTCTTCTCCTTCCTCT

## Data Availability

Not applicable.

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
