# Peer review of "(Pro)Renin Receptor Antagonism Attenuates High-Fat-Diet–Induced Hepatic Steatosis"

_biomolecules, 2023, doi:10.3390/biom13010142_

Round 1
Reviewer 1 Report
The study by Gayban AJB et al entitled “(Pro) Renin Receptor Antagonism Attenuates High-Fat-Diet–Induced Hepatic Steatosis” is well designed and written. However, there a few concerns/question which needs to be answered-
1. Were the PRO20 mediated effects on hepatic steatosis in HFD fed mice dependent on energy expenditure?
2. Authors only provided cumulative food intake data,
3. How many images/mice were analyzed for the represented data in Figure 1-3?
4. What was the effect of PRO20 on ALT & AST levels in the HFD-fed mouse model?
5. Please provide statistical sections including details about normality, outliers if any removed, and statistical methods used to derive the p values. In addition, please indicate the number of mice used for each study in respective Figure legends.
Author Response
- Were the PRO20 mediated effects on hepatic steatosis in HFD fed mice dependent on energy expenditure?
Response: We thank the reviewer for their comments. It is possible that PRO20-mediated effects on hepatic steatosis in HFD depend on energy expenditure. In the current study, we did not investigate energy expenditure but plan to perform these experiments in future studies.
- Authors only provided cumulative food intake data,
Response: In response to the reviewer’s comment, we have caloric intake as Figure 6C.
- How many images/mice were analyzed for the represented data in Figure 1-3?
Response: A total of 297 images were analyzed for the ORO data shown in Figure 1, 170 images for Figure 2, and 182 images for Figure 3. This works out to 5–10 images from each mouse, with 3–10 mice/group. We have added this information in the Methods section.
- What was the effect of PRO20 on ALT & AST levels in the HFD-fed mouse model?
Response: We did not measure ALT or AST in the HFD-fed mouse model because previous studies (Nakagawa et. Al, PMID: 25132496, Ito et. Al, PMID: 17300698) showed that a 60% kCal HFD only causes significant liver injury and inflammation when administered for longer periods of time (³30 weeks). Thus, the measurement of ALT and AST was not warranted, given the shorter duration of our HFD model.
- Please provide statistical sections including details about normality, outliers if any removed, and statistical methods used to derive the p values. In addition, please indicate the number of mice used for each study in respective Figure legends.
Response: We thank the review for their comment on rigor. We have added detailed information in the Statistical Analysis section and included animal numbers in the Figure legends.
Reviewer 2 Report
Thanks to the authors for an interesting article on the study of the treatment of non-alcoholic fatty liver disease. The introduction fully reflects the essence of the study, the methods are described in detail, the results are presented quite fully.
1. Why exactly after 2 weeks of diet the animals were given implants? Why were there no groups that were fed longer? Please explain why this particular period.
2. In the discussion section (lines 450-501), the paragraph looks more like an overview or part of an introduction, please rewrite it.
3. It seems to me that the article lacks the "Conclusion" section, which reflects the main conclusions and results of the study.
Author Response
- Why exactly after 2 weeks of diet the animals were given implants? Why were there no groups that were fed longer? Please explain why this particular period.
Response: Previous studies have shown that a high-fat diet induces hepatic steatosis after as few as 2 weeks. In a pilot study, we confirmed that 2 weeks of HFD also induced hepatic steatosis. Because our goal was to test whether PRO20 could reverse or slow steatosis development after pathology had already developed, similar to that in humans, we chose to start the treatment at 2 weeks after HFD. We have added a description of this rationale in the manuscript (Lines 235-237)
- In the discussion section (lines 450-501), the paragraph looks more like an overview or part of an introduction, please rewrite it.
Response: We thank the reviewer for their constructive comments for improving our paper and have rewritten the Discussion section to avoid repetition.
- It seems to me that the article lacks the "Conclusion" section, which reflects the main conclusions and results of the study.
Response: As suggested, we have added a “Conclusions” section at the end of the manuscript.
Reviewer 3 Report
I regret to advice that this manuscript seems to lack novelty because the main observations were already published. If the other referee holds different opinion regarding the judgment, do not hesitate to ask a new referee for a second opinion.
Author Response
I regret to advice that this manuscript seems to lack novelty because the main observations were already published. If the other referee holds different opinion regarding the judgment, do not hesitate to ask a new referee for a second opinion.
Response: We respectfully disagree. No study has been performed to investigate the effects of PRO20—which blocks activation of the PRR by prorenin/renin—in hepatic steatosis and fibrosis. This alone is important from a clinical standpoint because no FDA-approved therapeutics for the treatment of hepatic steatosis are currently available. Although there have been some reports on the effects of PRR genetic deletion on NAFLD, this approach is not feasible for application in humans. In addition, because the PRR itself mediates many signaling pathways that do not require the ligand, prorenin/renin, PRR deletion might act through different mechanisms than the PRR antagonist, PRO20. Indeed, this study found that PRO20 targets a novel pathway for hepatic lipogenesis.
Round 2
Reviewer 1 Report
The authors have satisfactorily answered my queries and incorporated suggested changes in the manuscript.
Reviewer 2 Report
I thank the authors for the changes made to the article.
Reviewer 3 Report
The findings are potentially interesting, but the contributions of PRR to liver fibrosis was well-known and easily reported about a promising therapeutic target for liver fibrosis in other journals. This manuscript seems to lack sufficient novelty for readers.